# Healthful vs. Unhealthful Plant-Based Restaurant Meals

**DOI:** 10.3390/nu17050742

**Published:** 2025-02-20

**Authors:** Kim A. Williams, Amy M. Horton, Rosella D. Baldridge, Mashaal Ikram

**Affiliations:** 1Department of Internal Medicine, School of Medicine, University of Louisville, 550 South Jackson Street, Louisville, KY 40202, USA; 2Office of Community Engagement, University of Louisville, Louisville, KY 40202, USA; 3Department of Medicine (Cardiology), University of Chicago—Endeavor Health System, 2650 Ridge Ave., Evanston, IL 60201, USA; mashaal.ikram12@gmail.com

**Keywords:** restaurants, vegan, vegetarian, healthful, unhealthful, refined grains, saturated fat, fried food

## Abstract

*Background:* Vegan/vegetarian (VEG) restaurants and VEG options in omnivore (OMNI) restaurants may serve unhealthful plant-based food that may be more harmful than a typical American diet. *Methods:* A sample of 561 restaurants with online menus were analyzed over a 3-year period. Each plant-based menu entrée was counted, up to a maximum of ten entrées per restaurant, meaning that a restaurant customer could select from ten or more healthful plant-based choices. Entrées containing refined grains (e.g., white rice and refined flour), saturated fat (e.g., palm oil and coconut oil), or deep-fried foods were counted as zero. *Results:* We evaluated 278 VEG and 283 OMNI restaurants. A full menu (10 or more plant-based entrées) was available in 59% of the VEG, but only 16% of the OMNI (*p* < 0.0001). Zero healthful options occurred in 27% of OMNI, but only 14% of VEG (*p* = 0.0002). The mean healthy entrée count for all restaurants was 3.2, meaning that, on average, there were only about three healthful plant-based choices of entrées on the menu, significantly more in VEG (4.0 vs. 2.4 *p* < 0.0001). The most common entrée reduction was for refined grains (e.g., white flour in veggie-burger buns or white rice in Asian entrées, n = 1408), followed by fried items (n = 768) and saturated fat (n = 318). VEG restaurants had a significantly higher frequency of adequate VEG options (≥7 options, 24% vs. 13%, *p* = 0.0005). *Conclusions:* Restaurants listed as VEG have a slightly higher number of healthful entrées than OMNI restaurants, which offer more limited vegan/vegetarian options. Given the published relationship between unhealthful dietary patterns, chronic illness, and mortality, we propose that detailed nutrition facts be publicly available for every restaurant.

## 1. Introduction

Plant-based eating patterns—often called vegan and/or vegetarian (VEG)—are growing in the United States and around the world. In the USA, the vegan food market reached approximately USD 11 billion as of 2023, with the meat-free and dairy alternative segments accounting for about USD 2.33 billion and USD 2.8 billion, respectively. Vegan market growth has been projected at 12% annually through 2030 [1].

This commercial growth has led to more VEG food options in grocery stores, an increase in dedicated VEG restaurants, and a growing number of VEG-labeled options in non-vegetarian (OMNI) restaurants. For example, more than 30,000 vegan and vegan-friendly restaurants are listed on platforms such as HappyCow [1]. Four main categories of VEG patrons fuel this increase: those driven by ethical concerns (anti-animal cruelty), environmental concerns (planetary sustainability), health-related reasons (reduced chronic disease risk), and/or religious beliefs. However, only a small minority—less than 2% of respondents in one National Health Interview Survey study—reported adopting a vegetarian or vegan diet for health reasons [2]. In fact, it has been estimated that 63% of the growth in plant-based eating is attributable to OMNI patrons rather than members of the VEG community.

Despite this growth, evidence suggests that some plant-based foods can be unhealthy [3,4]. For instance, juices and sweetened beverages, refined grains, potatoes/fries, and sweets may contribute to adverse health outcomes. Consumption of these items—and of animal foods that typically contain cholesterol and saturated fat—is recommended to be reduced or avoided according to the American College of Cardiology/American Heart Association (ACC/AHA) Guideline on the Primary Prevention of Cardiovascular Disease [5]. Similarly, the World Cancer Research Fund/American Institute for Cancer Research (WCRF/AICR) has compiled evidence linking fat and refined grain consumption to cancer risk [6], and the International Agency for Research on Cancer (IARC) has summarized data that associate the ingestion of fried foods and the resultant acrylamides with cancer production [7].

Many plant-based foods are designed and processed to appeal to popular tastes and may contain high levels of sodium, saturated fat, refined grains (such as white flour or white rice), and added sugar, or are deep fried. Consumption of these foods has been associated with increased mortality [8] and a higher risk of coronary heart disease—even when compared to a general American diet [3].

The primary purpose of this study was to determine how often a restaurant patron, seeking vegan options at an establishment advertised online as offering VEG choices, is presented with (a) a variety of healthful options (defined as 10 or more choices) and (b) options free from refined grains, saturated fat, or fried foods. A secondary aim was to assess the frequency with which detailed nutritional breakdowns are available online. Such information would allow consumers to identify unhealthy choices—those high in saturated fat or sodium or with a high sugar-to-fiber ratio.

## 2. Methods

In accordance with 45 CFR §46.102(f), this nutritional evaluation study was not submitted for institutional review-board approval or preregistered (https://clinicaltrials.gov), because it involved publicly available data and did not gather any patient data.

### 2.1. Study Design

Over a 3-year period, members of our assessment team visited a convenience sample of 561 restaurants purported to have vegan offerings, typically located close to the venue of a medical conference. This sample included 196 cities in 37 countries, mostly from USA (n = 373, in 26 states and the District of Columbia), Australia (n = 48), France (n = 29), England (n = 17), Germany (n = 16), Italy (n = 13), and Spain (n = 10) (see Appendix A for full list).

### 2.2. Inclusion and Exclusion Criteria

To qualify for inclusion in the sample, the restaurant was required to have an internet listing on a proprietary commercially available online resource, such as Caviar (https://www.trycaviar.com), DoorDash (https://www.doordash.com), Grubhub (https://www.grubhub.com), Happy Cow (https://www.happycow.net), search engines such as Google, or Uber Eats (https://www.ubereats.com/), and a confirmed online menu available for analysis containing a detailed listing of their non-dairy vegetarian (“vegan”) options. We excluded restaurants that only provided vegan options ad hoc, i.e., changing the menu items listed “upon request”, but are not listed online as having plant-based menu items. For each restaurant, we also searched online for the availability of nutrition facts, which would allow more in-depth evaluation of healthfulness, as mandated in the USA for any franchise with more than 20 locations [9].

### 2.3. Scoring of Healthy vs. Unhealthy Entrées

We analyzed the menu’s entrée items, rather than appetizers, side salads, side dishes, desserts, or beverages, compiling the name and analyzing the description of all items listed as “vegan”. Each entrée was given one point, up to a maximum of ten entrées per restaurant, typically using the top entry in each entrée section. For example, a VEG Chinese restaurant may have sections entitled the following: Vegan Meat Alternatives, Vegetable-Based Entrées, Noodle and Rice Dishes, Rice Dishes, and Tofu and Bean Curd Dishes; in this case of 5 sections, the first 2 entries in each category comprised the 10 selections for analysis.

Like one prior published analysis [3], an entrée was considered unhealthful and was deducted from the healthful entrée total if it contained either refined grains (e.g., white rice and refined flour), saturated fat (e.g., palm oil, coconut oil, coconut milk, or coconut cream), or deep-fried foods. The final restaurant healthy entrée total ranged from 0 to 10 (i.e., 10 entrée options with no removals for unhealthful ingredients). If an entrée with an unhealthy ingredient had a healthier option listed on the menu, e.g., brown rice instead of white rice, whole-grain bun vs. white-flour bun, or steamed tofu vs. fried tofu, the healthier option was credited to that entrée.

### 2.4. Refined Grains

Refined grains, i.e., processed to remove the germ (concentrated nutrients) and outer husk (bran fiber), leaving the endosperm (starchy middle), results in a higher glycemic index no longer balanced by fiber. Consumption of foods containing refined grains is associated with higher weight gain, dyslipidemia, cardiovascular disease, cancer, and mortality [6,10]. When noted, these were scored as −1. The presence of refined grains was determined by taste and texture, written description, or visual evidence in online images (e.g., baguette or a white bread veggie-burger or veggie-dog bun).

### 2.5. Saturated Fat

The presence of saturated fat was scored as −1 if the entrée listed cocoa butter, palm oil, coconut oil, or coconut milk, including both Impossible^TM^ meat (6 g per serving, https://www.fooddive.com/news/impossible-foods-reformulates-less-fat-than-beef/630756/ accessed on 1 February 2025) and Beyond Meat^TM^ (5 g per serving, https://www.beyondmeat.com/en-US/products/the-beyond-burger?variant=beyond-burger accessed on 1 February 2025). However, on 18 April 2024, Beyond Meat^TM^ announced a change from coconut oil to avocado oil, which reduced the saturated fat per serving to 2 g (https://www.beyondmeat.com/en-US/press/beyond-iv-the-fourth-generation-of-the-beyond-burger-and-beyond-beef-debuts-at-grocery-stores-across-the-u-s-including-at-walmart-and-kroger# accessed on 1 February 2025). Menu items entered after this date were not downgraded for saturated fat for Beyond Meat^TM^.

Vegan cheeses are generally at or below the recommended acceptable level of 10% of calories from 3.7 g of saturated fat [11], due to small serving size, given its use as a condiment (e.g., cheeseburger), or the use of low-saturated fat cheese (e.g., oat milk- or pea protein-derived cheese), which was usually not specified.

When identified on the menu, the nutritional components of other brands of plant-based meats were investigated to ensure a low saturated fat content. For example, LaVie bacon in Great Britain contains 1.3 g of saturated fat per 100 g serving, but 2000 mg of sodium (https://www.laviefoods.com/en/nutrition/ accessed on 1 February 2025).

### 2.6. Fried Foods

The health concerns surrounding the consumption of fried food center on its markedly increasing fat content and carcinogenic potential [12,13,14,15]. Given these concerns and epidemic planetary levels of obesity and metabolic syndrome, the presence of fried foods in an entrée was assigned a −1 score. However, for this study, stir-fried and pan-fried were not scored as deep-fried due to the lower oil content, but “battered” and “crispy” items were considered deep-fried, unless otherwise specified on the menu.

### 2.7. Statistics

Data on entrée counts and reductions are presented as mean and standard deviation. Comparison of means was performed with XLSTAT version 2024.1—Life Sciences, with the null hypothesis that there was no difference between the number of healthy options at VEG and OMNI restaurants. Normality of the distribution of entrée counts was evaluated with the Shapiro–Wilk test, and each group’s result was found to not follow a normal distribution. Therefore, the Mann–Whitney test was used to determine statistical significance of comparison of means of independent samples with a two-tailed *t*-test. Chi-square with one degree of freedom was utilized for frequency comparisons between VEG and OMNI restaurants. A *p*-value of < 0.05 was considered statistically significant.

## 3. Results

### 3.1. Restaurant Distribution

A total of 561 (278 VEG and 283 OMNI) restaurants were analyzed. In two restaurants, animal products were erroneously labeled as “vegan”: one VEG restaurant listed dairy (“non-vegan”) cheese on four items despite labeling as “vegan”, and one OMNI restaurant mistakenly labeled an entrée of “spicy stew of pork, brisket, turkey, veggies” as vegan on both the written and online menus. These menu items were excluded or censured from the analysis.

When the 373 United States of America (USA) and 188 non-USA restaurants were compared, there was no significant difference in the total available entrées per restaurant (USA = 6.6; non-USA = 6.2; *p* = 0.238), but the average number of healthful entrées trended slightly higher in the USA (3.4 vs. 2.9, *p* = 0.075).

Complete nutritional disclosure from online nutrition facts (including sodium content, calories, total and added sugar, or quantitation of total and saturated fat) was available in only 28 OMNI and 5 VEG restaurants (10% vs. 2%, *p* = 0.0001).

### 3.2. Entrée Healthfulness

There was an average of 4.5 healthful vegan entrées from OMNI restaurants (1276 out of a possible 2830 entrées), and 8.4 healthful entrées from VEG restaurants (2333 out of a possible 2780, *p* < 0.0001 vs. OMNI). A full menu (i.e., 10 or more plant-based-option entrées) was available in 59% of the VEG, but only 16% of the OMNI restaurants (*p* < 0.0001).

The mean healthfulness score of all restaurants was 3.2 ± 2.9, meaning that on average there were only about three healthful plant-based choices for entrées on the menu (Figure 1). VEG restaurants had slightly but significantly more healthful entrées than the OMNI restaurants (4.0 ± 2.9 vs. 2.4 ± 2.5, *p* < 0.0001). Out of 3609 entrées analyzed, there were only 1120 VEG and 676 OMNI healthful entrées (48% vs. 53%, *p* = 0.0998). VEG restaurants had the greatest number of healthy options, defined as a restaurant score of ≥7 out of the maximum 10, 24% vs. 13% for OMNI, *p* = 0.0005. A maximum entrée score of 10, i.e., no refined grains, no saturated-oil use, and no fried components, was found in only 2% of the restaurants. Zero healthful options were found in 26% (74/283) of OMNI, but only 14% (38/278) of VEG restaurants (*p* = 0.0002).

### 3.3. Score Reduction Analyses

A total of 2497 entrées had reduced scores, including 1408 reduced for refined grains, 321 for saturated fat, and 768 for fried foods. VEG restaurants had a higher frequency of score reductions than OMNI (1732 in 2333 entrées vs. 765 in 1276, *p* = 0.0001). As in Figure 2, when normalized for the number of entrées, score reduction for refined grains was the most common (e.g., non-whole-grain pasta, white flour in bread or burger buns, or white rice in Indian or Asian entrées) 40% for OMNI and 38% for VEG, *p* = NS. Reduction for saturated fat was more frequent in OMNI (12% vs. 7%, *p* = 0.0001), and reduction for fried items occurred more frequently in VEG (28% vs. 8%, *p* < 0.0001).

## 4. Discussion

This is the first published study to evaluate the healthfulness of plant-based restaurant offerings—adhering to the 2019 ACC/AHA Primary Prevention Guideline recommendations [5]—across all six continents where restaurants exist (i.e., excluding Antarctica). Our convenience sample of restaurants purported to serve plant-based (“vegan”) options revealed several central findings:There are many unhealthful plant-based foods labeled as “vegan” in both OMNI and VEG restaurants.Slightly more healthful options are available in VEG than in OMNI restaurants.Refined-grain use is common and represents the largest source of unhealthfulness in entrées.Healthful options are marginally more likely to be found in USA restaurants compared with non-USA restaurants.

Full disclosure of nutritional content—specifically sodium, sugar, and saturated fat—is not widely available. However, such disclosure is more frequent in OMNI restaurants than in VEG restaurants, largely due to the presence of plant-based options in USA restaurant chains that are mandated to disclose nutrition information.

These results are not unexpected, as the restaurants must respond to the desires of their customers. Customers are often limited by multiple barriers to healthy nutrition habits, which have been recently analyzed using artificial intelligence [16]. These included cultural and traditional habits; taste and familiarity; accessibility and affordability; nutritional-adequacy concerns; social norms and peer pressure; marketing and industry influences; lack of education; emotional attachments; convenience and time constraints; and psychological resistance to change. The authors pointed out that restaurants are “not in the healthcare business” and respond to market forces, since “they will not sell what we won’t buy”.

One previous, smaller publication on a similar topic [17] examined the menus of 73 restaurant chains with complete online nutrition information. Using the 2017 AHA criteria and a scoring system similar to ours (assigning either 0 or 1 point per entrée), they evaluated meals for the following criteria:Approximately 600 calories or fewer per meal;No more than 35% of calories from fat;Less than 5 g of saturated fat;Zero grams of trans fat;Cholesterol below 75 mg per meal;Sodium less than 700 mg per meal;At least 10 g of protein per meal;At least 5 g of fiber per meal.

They found that fewer than 20% of meals met the saturated fat and sodium criteria. In total, 22% of restaurant meals met zero to one AHA criteria, 50% met two to four criteria, 20% met five to six criteria, and only 8% met all seven criteria. They called for improvements in the healthiness of restaurant meals. However, that study was unable to evaluate meals using the more rigorous nutrition recommendations of the 2019 ACC/AHA Guidelines, and it did not assess fried foods or refined grains, which are known to accelerate chronic disease and cardiovascular risk.

In the United States, cardiovascular disease remains the leading cause of death. Nutrition-mediated acute and chronic diseases contribute significantly to rising healthcare costs, loss of life, and reduced economic and personal productivity [18]. Although the food and restaurant industries are not primarily in the healthcare business, their long-term sustainability depends on consumers’ purchasing habits. Consumer choices are heavily influenced by habit, culture, and television marketing of appealing yet unhealthy food—a phenomenon some have termed “marketing mortality” [19]. This influence disproportionately affects individuals with socioeconomic disadvantages [20,21]. For example, a recent study found that more than one-third of American families earning less than USD 9000 per year watched over five hours of television daily, compared with only 1% of families earning USD 150,000 per year [20]. Such media exposure has a particularly strong influence on economically disadvantaged children [21]. The percentage of children watching television for one hour or more per day increases from age 2 to 9 years, reaching 85% for children of low-educated mothers compared with 61% for children of highly educated mothers.

### 4.1. Healthful vs. Unhealthful Vegan/Vegetarian Food: Helpful or Unhelpful?

The recent ACC/AHA Cardiovascular Disease Primary Prevention Guidelines [5] recommend reducing consumption of foods that increase the risk of cardiac events and/or mortality. Observational data from the PURE trial indicate that, despite the risks associated with saturated fat, replacing refined carbohydrates with either saturated or unsaturated fats can reduce stroke and mortality [22]. Furthermore, a reduction in dietary sodium was shown to lower blood pressure and reduce cardiovascular events in the DASH trial and the Trials of Hypertension Prevention (TOHP) [23,24]. High consumption of sodium (>2000 mg daily), red meat (>14 g/d), sugar-sweetened beverages, and processed red meat has been associated with cardiovascular death and increased all-cause mortality in the NHANES [25].

A prospective cohort study of USA healthcare professionals [26] demonstrated that replacing animal protein with vegetable protein—thereby reducing the ingestion of compounds associated with increased cardiovascular and cancer risk, such as cholesterol, saturated fat, insulin-like growth factor (IGF-1), heme iron [27], advanced glycation end products (AGEs), polyaromatic hydrocarbons, and heterocyclic amines produced during high-temperature cooking [28]—is associated with reduced mortality. Additionally, precursors of trimethylamine-N-oxide (e.g., choline, betaine, carnitine, and phosphatidylcholine) have been linked to coronary artery disease, chronic kidney disease, and heart failure mortality; a plant-based diet may reduce these precursors through wholesale changes in the gastrointestinal microbiome [29]. Conversely, plant-protein consumption is associated with a 10% reduction in mortality for every 3% energy increment that replaces animal protein [26]. Moreover, consumption of sugar-sweetened and artificially sweetened beverages increases the risk of type 2 diabetes and cardiovascular events, with one daily serving linked to a 20% increase in diabetes risk [30]. Consumption of sugar exceeding 10% of daily calories has been associated with increased mortality [31].

The REGARDS (REasons for Geographic and Racial Differences in Stroke) study [32] found that the “Southern” dietary pattern—characterized by high intake of animal products and foods that scored negatively in our evaluation—is associated with a 56% higher risk of heart disease, a 50% increase in chronic kidney disease, and a 30% higher risk of stroke. Additionally, there is clear evidence that some plant-based foods are unhealthful [3]. Examples include fruit juices (which lack fiber), sugar- and artificially sweetened beverages, refined grains, potatoes/fries, and sweets. This pattern of eating may contribute to coronary disease development at a higher rate than the consumption of animal products [3].

### 4.2. Fried Food: Waste It or Just “Waist” It?

Deep frying a raw potato increases its fat content from 0.2 g per 100 g to 13.2 g—a 66-fold increase [10]. In cases where frying oil is rich in monounsaturated or polyunsaturated fats (e.g., olive oil or canola oil) and used in moderation, the patron’s lipid profile may improve [13]. However, for overweight or obese individuals, such frying may lead to weight gain and higher triglycerides. Enhanced taste from frying can also lead to increased food consumption due to dopamine release [33,34], thereby contributing to the rising global obesity rates [35,36] and systemic hypertension [37].

One isolated study reported that frying vegetables in olive oil is healthier than boiling them and may help prevent cancer, diabetes, and vision loss [38]. Frying may preserve vitamin C and B vitamins and could even increase the fiber content in potatoes by converting starch into resistant starch [39]. However, frying certain vegetables (such as in French fries or potato chips) may result in the formation of acrylamides via the Maillard reaction, which has been associated with cancer formation in animal models, though the evidence in humans is less clear [14,15]. Additionally, high-heat cooking forms AGEs, which are associated with inflammation, oxidative stress, and chronic diseases such as type 2 diabetes mellitus [40]. Finally, frying in repeatedly heated oils can produce trans fats, which are linked to adverse cardiovascular events [41].

### 4.3. Refined, but Against the Grain

We assigned negative scores to refined grains in our analysis based on their association with increased risk of death and disease compared to whole grains. A recent meta-analysis of 64 studies [10] demonstrated that whole-grain intake—with its high fiber content—is associated with a reduced risk of coronary heart disease; cardiovascular disease; various cancers; and overall mortality, including deaths from respiratory and infectious diseases, diabetes, and non-cardiovascular/non-cancer causes. The authors propose that replacing refined grains with whole grains could substantially reduce the burden of chronic disease if widely adopted. In contrast, international studies have indicated that consumption of refined grains is associated with increased total mortality, a higher incidence of major cardiovascular events, and elevated systolic blood pressure [8,42].

### 4.4. Saturated with Controversy

In 2014, the lay press widely reported that advice to avoid saturated fat was incorrect [43], challenging established science. However, subsequent reanalyses [44,45,46] and compilations of the data [47] have reinforced the link between saturated fat consumption and dyslipidemia, cardiac events, and mortality, despite some data suggesting that saturated fat is less harmful than refined carbohydrates [44]. Moreover, saturated fats can activate the inflammasome via toll-like receptor 4 (TLR4) signaling, leading to increased production of inflammatory mediators, whereas unsaturated fats do not exhibit this effect [48]. Higher intakes of saturated fat may also impair insulin signaling, contributing to insulin resistance, hepatic steatosis, obesity, and type 2 diabetes mellitus [49]. Diets high in saturated fats tend to be more energy dense and less satiating, which can promote weight gain [50]. In contrast, unsaturated fats—particularly omega-3 fatty acids—are neuroprotective and offer cognitive benefits relative to saturated fat intake [40]. There is also evidence that diets high in saturated fats disrupt the gut microbiome and increase gut permeability, while unsaturated fats may have beneficial effects on gut flora [37].

## 5. Limitations

Although this was an international study, most of the restaurants evaluated were in the USA and Westernized countries. This may limit the generalizability of our findings to restaurants in other regions. Our sampling was non-random and based primarily on proximity to the investigators’ homes, workplaces, and travel and urban medical meetings, which could introduce bias.

Moreover, our methodology did not permit personal sampling of every menu item. We relied on menu descriptions and available images to determine the use of refined grains. Similarly, assessing saturated fat content (specifically, whether it exceeded 10% of daily calories per serving) was often inferred rather than measured, given the limited availability of online nutritional information. In some cases, branded vegan products with known nutritional compositions—such as Impossible Meat™ (6 g per serving), Beyond Meat™ (5 g per serving), or Just Egg (0 g per serving)—were used in our grading. Notably, on 18 April 2024, Beyond Meat™ announced a switch from coconut oil to avocado oil, a change that should be associated with reduced LDL cholesterol and cardiac events [38]. Menu items containing Beyond Meat™ entered after this date were not downgraded for saturated fat. Lastly, due to incomplete nutritional information online, we could not quantitatively assess portion size or calories, levels of sodium, total fat, saturated fat, sugar, or sugar-to-fiber ratios.

## 6. Conclusions

In a very large international, although predominantly “Westernized”, evaluation of restaurants, healthful plant-based options are limited in OMNI restaurants but are slightly more available in VEG establishments, particularly in the US, compared with non-USA settings. Many menu items are high in refined grains—such as veggie-burger buns, white rice, and refined-grain pasta—and often include fried options and high levels of saturated fat, predominantly from coconut oil.

There is minimal transparency in the disclosure of nutritional facts that would help health-conscious patrons distinguish between healthful and unhealthful plant-based items. Since most restaurants do not provide detailed information on portion size, calories, sodium, total fat, saturated fat, total sugar, or added sugar content, even knowledgeable consumers may struggle to make informed choices. Given the well-established relationship between unhealthful dietary patterns, chronic illness, and mortality—and the relative ease with which nutritional information could be provided—we propose that detailed nutrition facts be made publicly available for every restaurant.

We recommend that the Food and Drug Administration in the USA (FDA) and international regulatory bodies expand requirements for nutritional disclosure beyond franchises with 20 or more locations. We suggest that restaurants reevaluate the healthfulness of their entrées by reducing the use of refined grains, excess sodium, saturated fat, excess sugar, and fried foods, using guideline-driven nutrition recipes and ingredients. Restaurants have the power, if not the responsibility, to promote health and sustainability rather than profits at the high cost of chronic disease and premature mortality.

## Figures and Tables

**Figure 1 nutrients-17-00742-f001:**
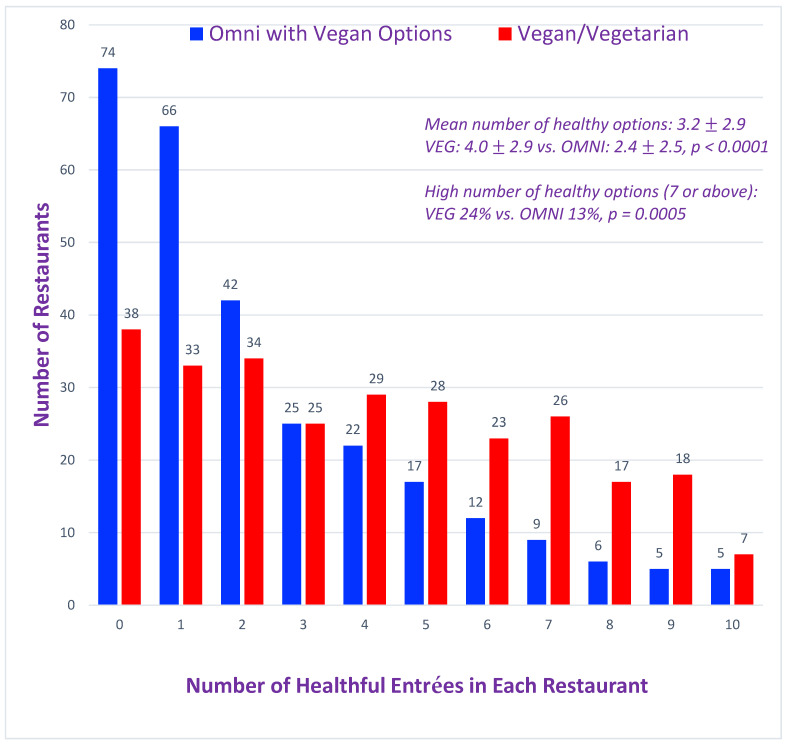
Frequency distribution of restaurant scores. Out of 561 restaurants, 278 were listed as vegan/vegetarian, and 283 were omnivore (OMNI) restaurants reporting to have vegan/vegetarian (VEG) options on the menu. The average healthfulness score was 3.2 ± 2.9, indicating just over 3 healthful plant-based options, defined as no animal products (i.e., zero cholesterol content), refined grains, excessive saturated fat or fried items. The OMNI restaurants scored slightly but significantly lower than VEG (*p* < 0.0001). There were more high-scoring (7 or above) VEG restaurants (*p* = 0.0005).

**Figure 2 nutrients-17-00742-f002:**
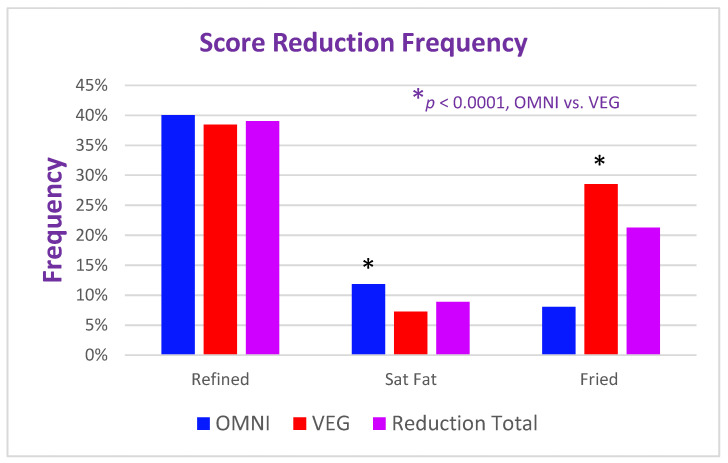
When normalized for the number of observations, 1276 for OMNI and 2333 for VEG, score reduction for refined grains (Refined) was the most common (e.g., non-whole-grain pasta, white flour in bread or burger buns, or white rice in Indian or Asian entrées) 40% for OMNI and 38% for VEG, *p* = NS. Reduction for saturated fat was more frequent in OMNI (12% vs. 7%, *p* = 0.0001) and reduction for fried items occurred more frequently in VEG (29% vs. 8%, *p* < 0.0001).

## Data Availability

Data are available in the Appendix A, without the individual performance of each restaurant.

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
