# Peer review of "Healthful vs. Unhealthful Plant-Based Restaurant Meals"

_nutrients, 2025, doi:10.3390/nu17050742_

Round 1
Reviewer 1 Report
Comments and Suggestions for Authors
This is an interesting study with adequate novelty. However, some points should be addressed.
- The Introduction section is quite short and it should be enriched by additional data. Especially the 3rd and 4rth paragraphs should be increased in length by adding more details for plant-based foods and diets and for unhealthy foods and dietary items connecting them with the increased risk for several chronic diseases beyong cardiovascular diseases.
- The financial data-information provided in the 1st paragraph should be specified whether are reffered to USA or all the world. Both of them are recommended to be included in ths 1st paragraph.
- More references are required to be included in the Introduction section.
- The 2nd paragraph of the Discussion section needs a bit more analysis.
- The sentence in lines 273-275 "The relationship is particularly influential on choices of economically disadvantaged children [18]." needs more analysis since it is reffred to an interesting topic.
- In the sections 4.1, 4.2, 4.3 and 4.4, the authors are mainly docused on the risk of the cardiovascular diseases. However, they should add more information for the increased risk of several other disorders such as metabolic disorders and obesity and cancer morbidity and mortality.
- English language editing is highly recommended especially in describing the results of the study. There are several grammar/syntax errors throughout the manuscript. In some cases, the language could be characterized as slang and not official. Several sentences are too long and they should be split into smaller sentences to be more easily readable.
- In general more references should be included with special focus on other diseases beyond cardiovascular disease.
Comments on the Quality of English LanguageEnglish language editing is highly recommended, There are several grammar/syntax errors throughout the manuscript. In some cases, the language could be characterized as slang and not official. Several sentences are too long and they should be split into smaller sentences to be more easily readable.
Author Response
Reviewer 1
This is an interesting study with adequate novelty. However, some points should be addressed.
- The Introduction section is quite short and it should be enriched by additional data. Especially the 3rd and 4rth paragraphs should be increased in length by adding more details for plant-based foods and diets and for unhealthy foods and dietary items connecting them with the increased risk for several chronic diseases beyong cardiovascular diseases.
Thanks. Although our intro was close to the 500-word limit recommended by journal editors, we have added statements and references on cancer development with refined grains and fried foods.
- The financial data-information provided in the 1st paragraph should be specified whether are reffered to USA or all the world. Both of them are recommended to be included in ths 1st paragraph.
Thanks. This has been added “(USA)” to the financial impact statement.
- More references are required to be included in the Introduction section.
Thanks. As above, we have added references 6 and 7.
- The 2nd paragraph of the Discussion section needs a bit more analysis.
We have expanded this discussion of the older study. However, we would appeal to use the original brief version, given that this study did not have the 2019 ACC/AHA prevention guideline nutrition section (3.1) which our corresponding author authored. Without analysis of the inclusion of fried food or refined grains, this study is out of date and misleading.
- The sentence in lines 273-275 "The relationship is particularly influential on choices of economically disadvantaged children [18]." needs more analysis since it is reffred to an interesting topic.
The authors are not certain what “reffred” means, but understanding that the reviewer preferred more information on reference 18, we have added the following text, “The incidence of children watching television ≥1 hour/day increased from age 2 to 9 years for all children, reaching 85% for children of low-educated mothers but only 61% for children of highly-educated mothers.”
- In the sections 4.1, 4.2, 4.3 and 4.4, the authors are mainly docused on the risk of the cardiovascular diseases. However, they should add more information for the increased risk of several other disorders such as metabolic disorders and obesity and cancer morbidity and mortality.
Thank you for this suggestion. These sections now cover all-cause mortality, stroke, hypertension, cardiovascular events, coronary artery disease, chronic kidney disease, heart failure mortality, type 2 diabetes, obesity, hypertriglyceridemia, cancer, respiratory diseases, infectious diseases, inflammation, hepatic steatosis, neurocognitive disease, and gut dysbiosis.
- English language editing is highly recommended especially in describing the results of the study. There are several grammar/syntax errors throughout the manuscript. In some cases, the language could be characterized as slang and not official. Several sentences are too long and they should be split into smaller sentences to be more easily readable.
Thank you, the manuscript has been expanded and reexamined as requested. Long sentences have been shortened. Please note that there is no “slang” in the manuscript, but the reviewer may be referring to the “catchy” titles and subtitles. As a long-term lecturer, research and guideline author and journal editor (KAW) and a professional scientific writer (AMH), we have learned that the catchy phrases increase reader interest, reader engagement, memorability and quotability. Please see the literature on this.
- Nielsen, J. (2006). F-Shaped Pattern For Reading Web Content. Nielsen Norman Group.
- Berger, J., & Milkman, K. L. (2012). What Makes Online Content Viral? Journal of Marketing Research, 49(2), 192–205.
- Letchford, A., Moat, H. S., & Preis, T. (2015). The Advantage of Short Paper Titles. Journal of Informetrics, 9(1), 284–295.
- Paivio, A. (1991). Dual Coding Theory: Retrospect and Current Status. Canadian Journal of Psychology, 45(3), 255–287.
- In general more references should be included with special focus on other diseases beyond cardiovascular disease.
Thank you. As above, these sections now cover all-cause mortality, stroke, hypertension, cardiovascular events, coronary artery disease, chronic kidney disease, heart failure mortality, type 2 diabetes, obesity, hypertriglyceridemia, cancer, respiratory diseases, infectious diseases, inflammation, hepatic steatosis, neurocognitive disease, and gut dysbiosis.
Reviewer 2 Report
Comments and Suggestions for Authors
Interesting idea of ​​this study, my recommendations are the following:
I recommend expanding the Introduction section regarding the size of comparative food portions on the two types of restaurants, their composition, the predisposition of the population/consumers and the influence on health of the two food typologies. In this part, only 5 bibliographic indexes are mentioned, which shows a poorly outlined substantiation, I recommend revising.
Method section – I recommend introducing a new section called Study design where the typology of the study and other specific aspects should be mentioned.
Section 2.2. I recommend rewriting the title – in the content you refer to the scoring scale, possibly mentioning the selection or another similar word.
Fig 1 I recommend mentioning what the vertical scale represents. Idem fig. 2.
Fig. 2 I recommend mentioning its title.
I recommend that the interpretation of figures 1 and 2 should not be in continuation of the titles.
At the end of the Discussion section, I recommend mentioning future research directions and the practical implications of the study.
Author Response
Response to Reviewer 2
Interesting idea of ​​this study, my recommendations are the following:
I recommend expanding the Introduction section regarding the size of comparative food portions on the two types of restaurants, their composition, the predisposition of the population/consumers and the influence on health of the two food typologies.
Thank you, these are excellent suggestions. However, they are VERY insightful discussion points, and have been placed in the Discussion section. We would note that the total “composition” issue was also placed in our Limitation section.
“we could not quantitatively assess portion size and calories, or levels of sodium . . . ”
We also added a section to the discussion on the “predisposition of population and consumers” along with a reference specifically about the barriers to healthful nutrition published a few days ago: Williams Sr. KA, Brown P. Chatting with ChatGPT: Barriers to Healthful Plant-Based Nutrition. ijdrp. 2025;7(1):8 pp. doi:10.22230/ijdrp.2025v7n1a531.
In this part, only 5 bibliographic indexes are mentioned, which shows a poorly outlined substantiation, I recommend revising.
Thanks. Although our intro was close to the 500-word limit recommended by journal editors, we have added statements and references on cancer development with refined grains and fried foods. The “substantiation” would typically review prior works in this field, but there are no such publications on the 2019 ACC/AHA nutrition recommendations.
Method section – I recommend introducing a new section called Study design where the typology of the study and other specific aspects should be mentioned.
Thanks. We have added the Study Design section.
Section 2.2. I recommend rewriting the title – in the content you refer to the scoring scale, possibly mentioning the selection or another similar word.
Due to the previous comment, this is now section 2.3. It is retitled to reflect the scoring scale.
Fig 1 I recommend mentioning what the vertical scale represents. Idem fig. 2.
Fig. 2 I recommend mentioning its title.
We agree with the reviewer. The original figure had clear titles which were requested to be removed by a previous Nutrients reviewer. We have added back the titles.
I recommend that the interpretation of figures 1 and 2 should not be in continuation of the titles.
This was also a request of other reviewers and we agree that the figure is better with a complete caption explaining the data so that it can stand alone when Nutrients is referenced by others.
At the end of the Discussion section, I recommend mentioning future research directions and the practical implications of the study.
We have added more wording to the end about future directions, but this is not about future research. This is definitive work – 561 restaurants – unlikely to be reproduced or continued. This is now about advocacy for health transparency of food in restaurants and designing menus to promote health rather than disease.
Round 2
Reviewer 1 Report
Comments and Suggestions for Authors
The authors have significantly improved their manuscript after revision process.
Author Response
February 9, 2025
RE: Manuscript Number: Nutrients-3465049
Dear Editors:
Please find enclosed the revised manuscript entitled, “Frequency of Healthful vs. Unhealthful Plant-Based Restaurant Meals.”
We appreciate the thoughtful reviews and have modified the manuscript accordingly. Our response to the academic editor is included here.
Sincerely,
|
Kim Allan Williams, MD, Sr., MACC, FAHA, MASNC |
Response to Academic Editor
- The title requires a change. The editor recommends that 'sickening servings!' should be replaced with a formal phrase.
The tag line has been removed, retitled “Healthful vs. Unhealthful Plant-Based Restaurant Meals”
- Abstract: The methodology of scoring food entries is not completely clear. The authors should state which food items would get the highest score and which items would score zero.
This has been clarified, as it now is a binary scoring system, i.e., not a scoring system, just a 1 or a 0 for each entrée. Thus, there is no “methodology of scoring” – all menu entrées containing beans, grains, nuts, seed, fruits, vegetables and mushrooms without animal products were counted, but that “count” was subtracted if it contained any ingredients categorized as refined grains, saturated fat or deep-fried food.
- The highest healthier choice of food is not currently clear in the methodology section of the summary. The limitations of the study should also clearly be stated with the conclusions (of the summary).
The authors assume the “summary” means “conclusions”? And that the “limitations” clearly stated in the prior paragraph should be repeated in the “conclusions”. We have repeated the geographic limitations of the very large study.
The “healthier choice of food” has been clarified throughout the manuscript. As above, there is no “healthier”, just healthful and unhealthful, with no scoring, just counting vegan entrées with or without unhealthful items “categorized as refined grains, saturated fat or deep-fried food”.
- Limitations: Please avoid mentioning restaurant names in the whole manuscript, such as lines 413-14, “Lord of the Fries”.
These lines have been removed.
All Figures (both Figure 1 and 2) should have numeric values appropriately, such as, mean with errors (like standard error mean or standard deviation). Alternatively, the values should be presented in tables.
The mean and standard deviations have been added to results and Figure 1, and removed from the figure legend. The academic editor will recall that Chi-Square frequency analyses (as in Figure 2) have no standard deviations.
Please follow the journal's guidelines to change the whole manuscript and references.
This has been accomplished.
Reviewer 2 Report
Comments and Suggestions for Authors
no comments
Author Response

(The authors gave the same response as above.)
